# The Latest Breakthroughs in Immunotherapy for Acute Myeloid Leukemia, with a Special Focus on NKG2D Ligands

**DOI:** 10.3390/ijms232415907

**Published:** 2022-12-14

**Authors:** Stefanie Maurer, Xiaoxuan Zhong, Betsy Deza Prada, John Mascarenhas, Lucas Ferrari de Andrade

**Affiliations:** 1Precision Immunology Institute, Department of Oncological Sciences, The Tisch Cancer Institute, Icahn School of Medicine at Mount Sinai, New York, NY 10029, USA; 2Division of Hematology/Oncology, Tisch Cancer Institute, Icahn School of Medicine at Mount Sinai, New York, NY 10029, USA

**Keywords:** AML, immunotherapy, NK cells, antibodies

## Abstract

Acute myeloid leukemia (AML) is a hematological malignancy characterized by clonal expansion of stem and myeloid progenitor cells. Immunotherapy has revolutionized the care for other cancers such as solid tumors and lymphomas, and has the potential to effectively treat AML. There has been substantial progress in the developments of immunotherapeutic approaches for AML over the last several years, including the development of antibodies that further increase the innate immunogenicity of leukemia cells by the inhibition of NKG2D ligand—particularly MICA and MICB—shedding, chimeric proteins such as IL-15 superagonist that expand natural killer (NK) cells, blockers of immunologic checkpoints such as NKG2A, and chemicals that indirectly increase expression of immune stimulatory proteins in leukemia stem cells. Furthermore, cellular therapies have been designed to enable alloreactive immunity by allogeneic NK cells or target leukemia antigens such as mutated NPM1. These immunotherapeutic approaches have demonstrated remarkable efficacies in preclinical studies and have successfully transitioned to early phase clinical trials, to establish safety and initial signal of clinical activity. Here, we briefly discuss some of the most recent and impactful developments in the AML immunotherapy field and provide our perspectives for the future directions of this exciting and new therapeutic opportunity.

## 1. Introduction

Leukemia is defined as cancer of the bone marrow that affects the production of blood cells. Acute myeloid leukemia (AML) is a type of blood cancer that specifically targets myeloid stem cells and its differentiation process. Under normal conditions, myeloid stem cells differentiate into three types of mature blood cells: platelets, red blood cells (RBC), and eventually white blood cells (WBC) [1]. Notably, AML affects the differentiation process, whereby myeloid stem cells exhibit impaired differentiation resulting in the accumulation of myeloblasts. Myeloblasts are immature WBC and, in the case of AML, accumulate in the bone marrow, outcompeting normal leukocytes and eventually exiting to the peripheral blood circulation and sometimes extramedullary sites [2,3]. Current treatment approaches for AML include cytotoxic chemotherapy, immune based therapies, epigenetic modifiers and small molecular inhibitors. As of recently, the focus has shifted onto immunotherapy which aims to stimulate the patient’s immune system to mount a response against the malignant cell population. By enhancing the immune response of pre-existing lymphocytes, such as natural killer (NK) cells and cytotoxic CD8 T cells, immunotherapy preferentially targets cancerous cells and minimizes damage to existing healthy cells, contrary to radiation and chemotherapy [4]. The scope of immunotherapy in research is immense, and here we will discuss some of the most recent strategies in translational development to treat AML.

## 2. Immunotherapeutic Strategies for AML

AML displays tremendous complexity with regard to expression of putative target molecules, differentiation status, proliferation and sensitivity to treatment. The potent ‘graft versus leukemia’ effect upon allogeneic hematopoietic cell transplantation (HSCT) utilized in high-risk disease lays ground to further the development and testing of immune-related therapies for AML. Several neoantigens result from recurrent chromosomal translocations in AML, e.g., PML-RARα, or gain-of-function mutations, e.g., FLT-ITD and NPM1, which renders them immunogenic target antigens [5,6,7,8]. Recent advances in personalized immunotherapy aim to establish expansion of tumor-reactive lymphocytes, particularly T cell clones. Rosenblatt and colleagues have reported a vaccine consisting of hybridoma cells derived from patient AML cells and autologous dendritic cells (DC) [9]. Vaccination with AML/DC fusion cells induced a strong T cell response in patients which prevented disease relapse in patients with residual disease. Another promising approach are drugs modulating the epigenome, e.g., hypomethylating agents (HMA), which have been approved for the treatment of AML patients not suitable for intensive chemotherapy. Notably, these compounds upregulate immunomodulatory molecules such as human leukocyte antigens (HLA) molecules and leukemia associated antigens including MAGE and PRAME [10,11,12]. 

### 2.1. Immune Checkpoint Inhibitors (ICI)

With ICI becoming a mainstay in immunotherapy across solid malignancies, the CTLA-4 inhibitor ipilimumab has been tested in patients with AML in a post-alloHSCT relapse setting. Upon treatment, infiltration and expansion of effector T cells in the peripheral blood was observed while activation of regulatory T cells was decreased. The responses have been promising, and in some cases lasted over a year [13]. Combination therapy of ipilimumab and decitabine has also pointed towards beneficial effects [14]. Other approaches have targeted the PD-1/PD-L1 axis with varying results. While blocking PD-1 using nivolumab or pembrolizumab demonstrated initial beneficial anti-leukemic activity in combination with azacytidine [15,16], PD-L1 inhibition using durvalumab in combination with azacitidine versus azacitidine alone proved to be safe but did not show superiority over single agent azacitidine [17]. Furthermore, multiple phase-1 and 2 clinical trials with AML revealed that ICI can generate severe immune-related adverse events, which were ultimately lethal in some cases. For example, anti-CTLA-4 administration following HSCT increased the incidence of graft-versus-host disease and was associated with death in a patient [13]. Anti-PD-1 in combination with azacitidine, a HMA, promoted grade-3 immune-related adverse events and, again, caused treatment-related deaths of two patients in a phase-2 trial [18]. A lower dosage of anti-PD-1 prevented treatment-related deaths of AML patients, but still led to grade-3 immune-related adverse events [19]. Response rates were promising (i.e., 59% for anti-CTLA-4, and 33% for anti-PD-1 in the high dosage trials), but due to the potential for immune related toxicity in AML patients, the use of checkpoint blockade has not moved forward for this leukemia [13,18,19]. As such, T-cell checkpoint blockade is yet not approved by the U.S. Food and Drug Administration (FDA) for the treatment of AML.

In addition to PD-1 and CTLA-4, there are other immune checkpoints that can be targeted to promote anti-AML immunity. For example, NKG2A is an inhibitory receptor on NK cells and CD8 T cells that recognizes HLA-E on tumor cells. NKG2A inhibits the cytotoxic functions of NK cells and CD8 T cells, and has emerged as a new immune checkpoint in cancer. Monalizumab, an anti-human NKG2A antibody, performed extremely well in an early phase clinical trial in squamous cell carcinoma patients [20]. Monalizumab also generated promising therapeutic effects in pre-clinical research studies in AML models by promoting NK cell-mediated killing of patient-derived AML cells [21]. Infusion of NKG2A+ NK cells together with human primary leukemia cells in immunodeficient mice rescued them from leukemia onset when NK cells were pre-treated with anti-human NKG2A antibody. In a phase-I study with fifteen patients with hematological malignancies (AML n = 9, myelodysplastic syndromes n = 3, lymphoma n = 1, chronic lymphocytic leukemia n = 1, and myelofibrosis n = 1), monalizumab was administered post-alloHSCT without dose limiting toxicities. Although one patient died from complications of graft-versus-host-disease, thirteen patients maintained complete remission (CR) without evidence of clinically significant gra.ft-versus-host-disease after a median follow up of 22 months [22]. Therefore, NKG2A blockade is an alternative ICI that holds strong promise for AML in the post HSCT maintenance setting. TIM-3, an inhibitory receptor that regulates adaptive and innate immunity, is expressed on both, immune cells and leukemic stem cells/blasts, but not normal hematopoietic stem cells. This TIM-3 a promising target in AML/MDS as it may restore immune effector function and simultaneously target LSC/leukemic blasts [23]. Sabatolimab (MBG453), a humanized IgG4 antibody targeting TIM-3, was tested in a phase Ib dose-escalation study in combination with HMA and was not only tolerated in patients with AML or MD but also showed encouraging anti-leukemic activity (NCT03066648) [24]. Another promising ICI which is being investigated in AML is CD27 which is found on lymphocytes and hematopoietic stem cells. It binds to its ligand CD70 on activated lymphocytes and dendritic cells. Due its role in leukemia stem cell expansion and effector function of cytotoxic lymphocytes, blocking of the CD27-CD70 axis may constitute an effective immunotherapeutic strategy. Since treatment with HMA results in upregulation of CD70 on leukemic stem cells from AML patients, a phase I clinical trial studied combination of HMA with the monoclonal anti-CD70 antibody cusatuzumab (NCT03030612) [25]. Blockade of CD27-CD70-interaction using cusatuzumab also increased NK-cell-mediated cytotoxicity and reduced leukemic stem cells in AML patients [25]. This promising strategy is now evaluated in a follow-up phase 2 AML2001 CULMINATE study (NCT04023526). New ICI targets might constitute promising strategies. Given the rather limited success of ICI as monotherapy in such a complex disease as AML, efforts are being made to test combination of either different ICI or ICI + HMA with regard to safety and potential synergistic effects [26]. Results of combination therapies are reviewed elsewhere. 

### 2.2. T and Natural Killer-Based Cellular Therapies

HSCT constitutes one of the early immune-based treatment approaches for leukemia. AML remains the most frequent indication for allogeneic HSCT [27]. The resulting graft-versus-leukemia response has shown optimal efficacy against AML at the time of first CR and increasingly evaluated in the setting of minimal residual disease (MRD) [28,29]. NK cells are recognized as key modifiers in this graft-versus-leukemia effect. They express a variety of inhibitory receptors (KIR) which bind to “self” major histocompatibility complex (MHC) class I epitopes. This mechanism prevents killing of the host cells in the absence of cellular stress. Donor NK cells in haploytype-mismatched HSCT, however, are not inhibited by the host cells since they do not express “self” MHC class I epitopes. This missing self phenotype of leukemic cells in an alloHSCT setting largely contributes to the efficacy of the latter. Meanwhile infusion of allogeneic NK cells in AML patients has shown to be safe and feasible and may contribute to disease control [30]. There are also successful approaches using transfer of autologous NK cells for immunotherapy of AML. Fehninger and colleagues reported on the transfer of haploidentical NK cells which had been differentiated into memory-like NK cells using a cytokine cocktail consisting of interleukin (IL)-12, IL-15, and IL-18 (NCT02782546) [31]. Fifteen high-risk patients underwent reduced-intensity conditioning followed by haplo-HSCT, and infusion of same-donor memory-like NK cells at day 7 together with an IL-15 superagonist which was given over 3 weeks. Although 87% of patients reached composite CR after 28 days and functional NK cells expanded in vivo, the median event-free survival was only 3.2 months. Although the efficacy of this regimen was disappointing, the approach is highly promising and requires further optimization. The IL-15 superagonist N-803 used in this study holds promises for successful expansion of endogenous cytotoxic lymphocytes or lymphocyte infusions in various malignanies. N-803 comprises a mutated version of IL-15 which is bound to a dimeric IL-15Rα-IgG1 Fc chimera and displays enhanced serum half life and biologic activity, compared to recombinant IL-15 [32].

Adoptive transfer of chimeric antigen receptor (CAR) T cells is increasingly pursued to treat several malignancies. Due to the lack of suitable, leukemia-restricted target antigens, limited clinical progress has been made with CAR-T cell therapy in the context of AML. Currently, CD33 is among the most investigated targets in AML and clinical trials testing CAR-T cells specific for this antigen have been evaluated or are ongoing (e.g., NCT03126864, NCT04835519). So far, there is only one case reported where a patient with refractory AML showed cytokine release syndrom (CRS) followed by a transient reduction of leukemic burden in the bone marrow [33]. To enhance treatment benefit, dual CAR-T cell trials which target two AML antigens in parallel, e.g., CD33-CLL1, are recruiting (NCT03795779, NCT05248685). Albeit, one of the biggest challenges in designing this immunotherapeutic regimen is to balance treatment efficacy and management of immune related adverse effects. While development of mild CRS is desirable, as it is indicative of an active T cell immune response, severe CRS is among the reasons for fatal outcomes in particular for T cell related immunotherapies. 

NK cells can also be engineered to express a CAR, and CD19-targeted CAR-NK cells were associated with 73% response rate without CRS or neurotoxicity in patients with CD19+ lymphoma [34]. CD33 specific CAR-NK cells are being tested in AML (NCT05008575). However, a caveat is that CD33 is also expressed by healthy myeloid cells and CD33-targeted CAR-T cells were associated with severe hepatotoxicity attributed to likely immune reaction against Kupffer cells [35]. In addition to CD33, CD123 (Il-3R α) is being evaluated as target antigen for CAR-T cell-based approaches. The CD123 CAR MB-102 was granted the status of orphan drug designation by the FDA following promising results from their phase I trial (NCT02159495) where relapsed refractory AML patients who had received prior HSCT attained CR without significant CRS, and other CD123 CAR T cells are currently tested (NCT04318678, NCT04265963) [29]. Another leukemia specific target antigen employed for cellular therapies is mutated NPM1. MHC class I ligandome analyses of patient-derived AML samples identified several peptides derived from mutated NPM1 that bind to common HLA types. A single chain variable fragment (scFv) was generated which specifically binds to the complex of mutated NPM1:HLA-A2 and was employed to generate CAR-T cells specific for mutated NPM1 [36]. Recent work also reports on the generation of CAR-NK cells bearing the aforementioned scFv [37]. Both, CAR-T as well as CAR-NK cells displayed anti-tumor reactivity towards NPM1-mutated AML. Collectively, both T cells and NK cells have strong anti-leukemic potential as cellular therapies, with the latter ones having the potential advantage of less CRS related toxicity.

### 2.3. Antibody-Based Approaches

First generation antibodies used in AML aim to stimulate immune effector cells by their Fc domain, which induces NK cell antibody-dependent cellular cytotoxicity (ADCC), antibody-dependent cell phagocytosis (ADP), or complement-dependent cytotoxicity (CDC). Therapeutic antibodies may further be conjugated to toxic mojeties, i.e., antibody-drug conjugates (ADC) or radioisotope conjugates (RIC). A newer development is to enhance the anti-leukemic effects of T or NK lymphocytes by bi-/multi-specific antibody formats. A variety of the first class of naked antibodies has been studied in clinical trials. Most prominent example is lintuzumab which targets CD33 and was given alone or in combination with cytotoxic agents, the efficacy however was limited while toxicity was observed in patients within the treatment groups [38]. An ADC of lintuzumab has been tested in the clinic but was associated with limited efficacy (NCT00038051) while its RIC version is still under evaluation (NCT03441048). Additionally, several clinical trials are currently targeting CD33 in bi-/multispecific antibody formats (e.g., NCT05077423) [29]. Other studies targeting CD44 or CD123 have demonstrated limited clinical benefit for patients with AML (NCT01641250, NCT02848248) [29]. On the contrary, the anti-CD47 “don’t eat me” antibody magrolimab was associated with promising responses in a phase 1b trial (NCT03248479) where combination therapy of magrolimab and azacitidine demonstrated durable responses and a benefit in overall survival in a cohort of patients with treatment naïve AML harboring TP53-mutated AML who were unsuitable for intensive chemotherapy [39]. 

## 3. Targeting NKG2D Ligands 

Malignant transformation often induces the expression of ligands for the NKG2D receptor, which promotes the cytotoxic functions of NK cells and CD8^+^ T cells [40,41]. NKG2D ligands (NKG2DL) are commonly absent on healthy cells, but upregulated by various types of cancers including AML [42,43]. The NKG2D ligandome is comprised of MHC class I chain-related protein A and B (MICA, MICB), and the UL16-binding proteins 1–6 (ULBP1-6) in humans. The high prevalence of expression on malignant cells and the predominantly tumor-restricted expression make NKG2DL promising targets for cell-based or antibody-based immunotherapeutic strategies, which are reviewed below.

### 3.1. NKG2D Receptor-Based Cellular Therapies

CAR-T cells have also been developed to target the NKG2D ligandome as a whole (Table 1). To this end, a CAR construct consisting of the natural full-length human receptor NKG2D and the intracellular domain of CD3ζ was developed which binds to all 8 ligands (Figure 1a). Sallmann and colleagues reported that adoptive transfer of the NKG2D specific CAR (CYAD-01) T cells in patients with AML or MDS showed promising response rates of 46% in a standalone phase I clinical trial enrolling relapsed/refractory AML patients (NCT03018405) [44]. No major adverse events have been observed. The same CAR is being evaluated in AML and MDS patients receiving prior lymphodepletion [45]. However, another trial revealed that chimeric NKG2D receptor T cells were safe in AML patients but did not generate clinical response [46]. The inconsistency in such clinical outcomes may be caused, at least in part, by the regulation of NKG2DL via epigenetic and post-translational mechanisms that are reviewed below. Therefore, cellular therapies that target the NKG2D pathway may be best combined with other approaches that enhance NKG2DL expression in leukemia cells. The strategy to use receptor-based cellular therapies appears elegant as they target all known and potentially unknown NKG2DL. However, the affinity of physiologic receptors such as NKG2D to its ligands is far lower than that of therapeutic antibodies which is expected to lead to reduced efficacy of receptor-based therapies. While for example the therapeutic antibody trastuzumab was reported to have an affinity of 4.0 × 10^−10^ M (equilibrium dissociation constant, Kd) towards its target HER2 [47], binding affinity of MICA/B to their cognate receptor is at least 2000 fold lower (highest reported affinity  8 × 10^−7^ M for MICB) [42]. It will need to be investigated whether enhanced avidity (binding of several NKG2D-NKG2DL pairs between two cells) could compensate for the above. Another important aspect with regard to receptor-based cell therapies may be that soluble NKG2DL are reported to downregulate endogenous NKG2DL. It needs to be investigated whether this may also influence surface level of NKG2D-receptor engineered cells.

### 3.2. Drug-Mediated Increase of NKG2DL Gene Expression

AML cells often have aberrant epigenetic modifications that inhibit NKG2DL expression. For example, Paczulla and colleagues demonstrated that leukemia stem cells lack NKG2DL expression and consequently escape NK cell recognition [48]. Inhibition of Poly ADP-ribose polymerase 1 (PARP1) by either small interfering RNA or the PARP1 inhibitor AG-14361 results in upregulation of NKG2DL on the surface of patient-derived AML cells. Inhibition of PARP1 sensitizes AML cells to NK cell immunosurveilance in vivo. It has also been shown that higher methylation levels of MICA, ULBP1/2 was associated with absence of NKG2DL surface expression in AML. Treatment with an HMA could restore NKG2DL expression level and thus immune recognition [49]. We and others also demonstrated that different histone deacetylase (HDAC) inhibitors like the FDA-approved drugs romidepsin or valproic acid upregulate NKG2D ligands (MICA/MICB) on the surface of AML cells, thus providing further immunoregulatory signals and sensitizing them to NKG2DL-based immunotherapy [50,51] (Table 1). Finally, HMA exposure have also been shown to increase NKG2DL expression in leukemia cells, which in turn are recognized and killed by NK cells (Figure 1b) [49]. Among the various small molecules identified to upregulate NKG2DL expression are many epigenetic regulators which may have a plethora of other effects in vivo. Thus it needs to be elucidated whether and how they i) modulate expression of other immunoregulatory molecules on malignant and benign cells, ii) how this affects the overall immunogenicity of the malignant cells and iii) whether these compounds reduce activation of immune cells.

### 3.3. ADCC and Antibody-Mediated Inhibition of MICA and MICB Shedding

NKG2DL display highly variable expression pattern within different tumor entities and among individual AML patients [43]. Strategies to employ NKG2D-Ig fusion proteins which simultaneously target all NKG2DL have been introduced. Fc-domain optimization further enhanced their potential to induce NK cell mediated ADCC against AML cells [52]. This strategy to sensitize immune cells to NKG2DL bearing leukemic cells was expanded by Maerklin and colleagues who reported on NKG2D-antiCD3/antiCD16 fusion proteins which induced T cell and NK cell reactivity, respectively [53]. As mentioned above in the context of cellular therapies, using NKG2D-based antibody constructs has the limitation of reduced affinity of the receptor-based therapeutic to its target and it needs to be investigated with different antibody-formats, whether this lack of affinity could be compensated by enhanced avidity. However, cancer cells escape immunosurveillance among other pathways through proteolytic cleavage of MICA/B molecules from the cell surface. Shedding of MICA/B is mediated by endoplasmic reticulum protein 5 (ERp5) and several A Disintegrin and Metalloproteinases (ADAM) family proteins and matrix metalloproteinases (MMP) family proteins [54,55,56]. ERp5 binds to the conserved motif in the α3-domain of MICA and MICB [54]. This binding process induces a conformational change of MICA/B to allow further proteolytic cleavage. Preclinical work has demonstrated the efficacy of a vaccine targeting the α3-domain of MICA/B in order to prevent shedding and thus reinforce NK and T cell immunosurveillance in solid tumors [57]. Future studies will be needed to validate whether this elegant strategy could also benefit patients with AML (Table 1). 

We have developed an antibody (7C6) that specifically binds the α3-domain of MICA/B and thereby prevents shedding. The antibody does not block interaction with NKG2D, which occurs in the MICA/B alpha-1 and alpha-2 domains. In a solid tumor-focused study, we have demonstrated that tumor cells are killed by NK cells in a NKG2D-dependent manner when treated with 7C6, thus indicating that inhibition of MICA/B shedding promotes NKG2D-mediated recognition of malignant cells by NK cells. Interestingly, the antibody is also recognized by Fc receptors that, in turn, trigger ADCC. In in vivo melanoma models, mice treated with 7C6 have fewer metastases in the lungs and such therapeutic effect is mediated by NK cells [58]. Melanoma metastases with mutations that confer resistance to CD8 T cell-driven immunity are also inhibited by 7C6, in a NK cell-mediated specific manner [59]. 7C6 due to its dual mode of action constitutes a promising immunotherapeutic strategy for the treatment of AML. This is corroborated by the fact that a clinical grade MICA/B α3-domain antibody is currently tested in a clinical trial in advanced solid tumors (NCT05117476). Therefore, antibody-mediated inhibition of MICA/B shedding promotes NK cell-driven anti-tumor immunity and has immunotherapeutic potential. Comparing to the above mentioned challenges of NKG2D-receptor-based therapies, the considerations for specific targeting of MICA/B are contrary. While there is high affinity binding to the tumor antigens MICA/MICB, not all leukemic cells express ligands of the MIC family and cannot be reached by this targeted immunotherapy.

Given the ability of the MICA/B antibody to promote anti-tumor immunity, we have recently applied 7C6 to pre-clinical models of AML. In two mouse AML models, 7C6 inhibits the leukemia outgrowth in the blood and bone marrow—in one of the models the reduction was of 10-fold while in the other model 100-fold, thus highlighting the outstanding immunotherapeutic potential of this approach. Surprisingly, in large mechanistic experiments where different leukocyte populations were depleted (e.g., myeloid cells, and innate and adaptive lymphocytes), we discovered that 7C6 inhibits AML primarily by ADP by macrophages. Although we detected contribution by NK cells for the therapeutic efficacy of 7C6, macrophage depletion completely abroagted its efficacy against AML (Figure 1c) [50]. Therefore, the MICA/B antibody is capable of triggering multiple arms of immunity.

Human cancers regulate MICA/B not only by a post-translational modification (i.e., cleavage) but also by an epigenetic mechanism. Many HDAC inhibitors are known to induce MICA/B expression [51,60,61,62]. We took advantage of this knowledge by screening a select group of narrow spectrum HDAC inhibitors, and discovered that romidepsin induces MICB mRNA expression in human AML cells [50]. Such finding extends previous study that demonstrated that romidepsin induces increase of MICA/B expression in other hematological malignancies, primarily lymphoma [63]. Of importance, the uniqueness of our approach is the combination of romidepsin with 7C6, which is the antibody that inhibits the shedding. Romidepsin + 7C6 induces high levels of surface MICA/B expression in human AML cells, by mechanisms that involves induction of MICA/B gene expression followed by the stabilization of the translated proteins on the cell surface by romidepsin and 7C6, respectively. Such AML cells are phagocytosed by macrophages, as consequence of Fc receptor binding, and 7C6 + romidepsin inhibits human AML cell growth in vivo [50]. Therefore, an immuontherapeutic strategy for AML can be achieved by the mechanism-driven combination of the MICA/B antibody with romidepsin in pre-clinical models.

**Table 1 ijms-23-15907-t001:** Immunotherapies targeting NKG2DL in AML.

	Preclinical Data	Clinical Data
**NKG2D-based cellular therapies**		
NKG2D CAR-T cells		NCT03466320 NCT02203825 NCT04658004 NCT03018405
NKG2D CAR-NK cells		NCT05247957 NCT04623944
**NKG2DL induction**		
PARP1 inhibition	siRNA, AG-14361 [48]	NCT05319249
HDAC inhibition	Romidepsin [50]	
	Valproic acid [51,64]	
HMA	5azaC and DAC [49]	
other	all-trans-retinoic acid [64]	
**Ab targeting of NKG2DL**		
ADCC induction	NKG2D-Fc-ADCC [52]	
bispecific	NKG2D-CD3/CD16 [53]	
shedding inhibition	[50]	

## 4. Conclusions

Immunotherapy has demonstrated remarkable efficacy for several cancers and holds potential promise for the treatment of AML. Given the heterogeneity of AML combined with the dynamics of clonal expansion during the course of the disease, we reason that a treatment regimen consisting of at least two agents is most likely to improve upon standard of care, e.g., systemic chemotherapy combined with immunotherapy such as leukemia directed antibodies or immune checkpoint inhibitors. However, further studies are warranted to test the efficacy and safety of combination therapies in different clinical settings of AML before patients can benefit from these novel and highly promising therapeutic regimen. Furthermore, in addition to the treatments reviewed above, other immunotherapeutic strategies are under late stage development, such as antibody-mediated blockade of SIRPA or CD47 that represent “do not eat me” signals for inhibition of macrophage-mediated immunity [39,65,66,67]. Collectively, recent advances have indicated the promising potential of immunotherapy for AML and with a spotlight on modulating innate immunity in AML immunotherapy research.

## 5. Patents

L.F.d.A. is co-inventor in an issued patent about an alpha-3 domain-specific antibody.

## Figures and Tables

**Figure 1 ijms-23-15907-f001:**
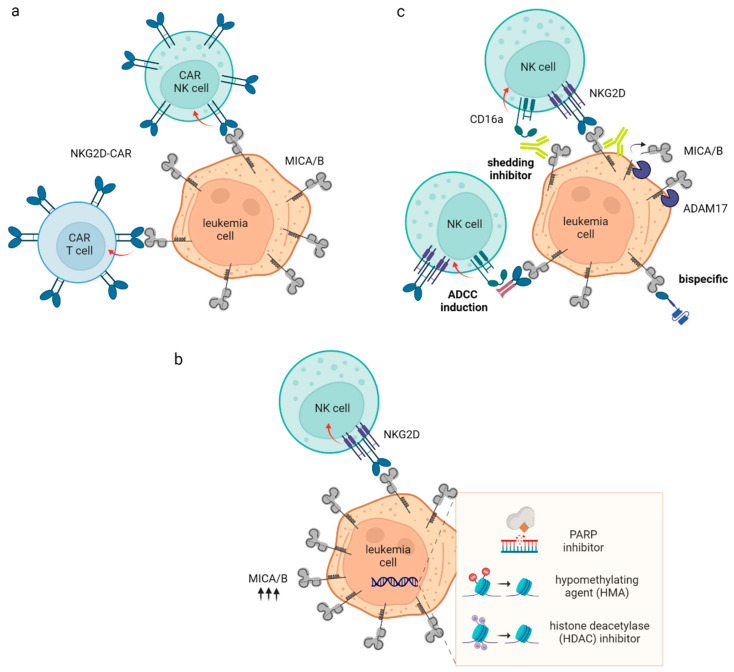
Modalities of NKG2DL-based immunotherapy in AML. (**a**) Cellular therapies consisting of NK or T cells expressing the NKG2D ectodomain fused to cytoplasmic signaling domain(s). NKG2D-CAR recognize the whole NKG2D ligandome and induces potent T and NK cell antitumor reactivity, respectively. (**b**) Epigenetic compounds such as PARP/histone deacetylase (HDAC) inhibitors or hypomethylating agents (HMA) lead to substantial upregulation of surface NKG2DL expression and thus re-sensitize AML cells to NKG2D-mediated NK/T cell immunity or make them susceptible to NKG2DL-based targeted therapies. (**c**) Antibodies (Ab) and engineered antibody-like formats which bind to NKG2DL and engage CD16a on NK cells. An Ab which binds to the α-3 domain of MICA/B and thereby blocks their shedding from the tumor surface is currently evaluated in a clinical trial. Additionally, Fc-optimized NKG2D-Ig fusion proteins have been tested to induce ADCC against target cells which express any of the 8 NKG2DL and enhanced the anti-leukemic reactivity of NK cells. Similarly, an NKG2DxCD16a bispecific has been tested which consists of the NKG2D ectodomain fused to an anti-CD16a single chain variable fragment (scFv) and consequently displays higher affinity to the Fc receptor than Fc domains. This figure was created with BioRender.com.

## Data Availability

Not applicable.

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
