# Peer review of "The Latest Breakthroughs in Immunotherapy for Acute Myeloid Leukemia, with a Special Focus on NKG2D Ligands"

_ijms, 2022, doi:10.3390/ijms232415907_

Round 1

Reviewer 1 Report

Dr Maurer et collab described in this review The Latest Breakthroughs in Immunotherapy for Acute Myeloid Leukemia, with a special focus on NKG2D ligands.

This is very well constructed as review based on most of relevant clinical trials integrating most recent and impactful developments in the AML immunotherapy field and provide some perspectives for the future directions this exciting and new therapeutic opportunity reported as practical and personalized adapted treatment choices.

The manuscript follows mainly options used in lasts years in most clinical trials and allogeneic HSTC in AML, including Hypomethylating agents and FLT3 inhibitors based on most relevant studies  also  addition to PD-1 and CTLA-4, there are other immune checkpoints that can be based on significant published data, targeted to promote anti-AML immunity.

The authors described very clearly the crucial aspects in post-immunotherapy or transplant setting as immunological allogeneic status with so important maintenance balance between GVL and GVHD. Adoptive transfer of chimeric antigen receptor CAR -T or CAR-NK cells is increasingly used to treat several cancers , but was very well highlighted by the authors  the limit of clinical progress  with CAR-T cell therapy in the context of AML due to the lack of suitable, leukemia-restricted target antigens.

Anyway, the high prevalence of expression on malignant cells and the predominantly tumor-restricted expression make NKG2DL promising targets for cell-based or antibody-based immunotherapeutic strategies, which are reviewed very clearly by the authors in this paper.

Author Response

We thank the reviewer for the helpful suggestions. Please find our responses attached.

Reviewer 2 Report

The article is discussing a recent complicated subject and novel  new ideas more explanation may be needed. Yet the article in it's form can be accepted as concised  review.

Author Response

We thank the reviewer for their insightful comments about further discussions of the complicated subject.

Reviewer 3 Report

The authors reviewed the Latest Breakthroughs in Immunotherapy for Acute Myeloid Leukemia (AML), with a special focus on NKG2D ligands including their own research. The manuscript is intriguing for the readers of “International Journal of Molecular Sciences” because Immunotherapy for AML is one of the major topics for hematological malignancies. However, there are several issues to be clarified.

Majors)

In the item of “3. Targeting NKG2D ligands”, the authors described 3 promising modalities of NKG2DL-based immunotherapy in AML as shown in Figure 1.

It is better to describe limitation of each modality at least in brief.

Minors)

1) In Abstract, MICA and MICB should be described as NKG2D ligand.

2) In Abstract or in Text, MICA and MICB should be spelled out as ” MHC class I chain-related gene A” and “MHC class I chain-related gene B”, respectively.

3) In Figure legend, Figure 2 should be Figure 1.

4) In the figure of Figure 1c, is blue cell in bottom T cell, NK cell or other cell?

Author Response

We thank the reviewer for their insightful comments and suggestions. Please find attached our reply.
